# Pharmacological Agents with Antiviral Activity against Monkeypox Infection

**DOI:** 10.3390/ijms232415941

**Published:** 2022-12-14

**Authors:** Andrea Zovi, Francesco Ferrara, Roberto Langella, Antonio Vitiello

**Affiliations:** 1Ministry of Health, Viale Giorgio Ribotta 5, 00144 Rome, Italy; 2Asl Napoli 3 Sud, Pharmaceutical Department, Dell’amicizia Street 22, 80035 Nola, Italy; 3Italian Society of Hospital Pharmacy (SIFO), SIFO Secretariat of the Lombardy Region, Via Carlo Farini 81, 20159 Milan, Italy

**Keywords:** monkeypox, *Orthopoxvirus*, pharmacology, antivirals

## Abstract

Monkeypox infection is caused by a virus of the genus *Orthopoxvirus*, a member of the *Poxviridae* family. Monkeypox virus is transmitted from individual to individual through contact with lesions, body fluids, and respiratory droplets. The infection caused by monkeypox is usually a self-limited disease with mild symptoms lasting 2 to 4 weeks. Monkeypox typically presents with fever, rash, and enlarged lymph nodes. New vaccines have recently been authorized for the prevention of monkeypox infection, whereas there are no specific pharmacological antiviral treatments for monkeypox infection. However, because the viruses which cause adult smallpox and monkeypox are similar, antiviral drugs developed in the past have also shown efficacy against monkeypox. In this review, we highlight the in vitro and clinical evidence found in the literature on the efficacy and safety of pharmacological agents with antiviral activity against monkeypox infection and the different regulatory aspects of countries.

## 1. Introduction

### Smallpox and Monkeypox Infections

Smallpox viruses belong to a very large group of DNA viruses that can infect both animals and humans. Smallpox originated with the infection of cows, a subspecies of vertebrates, and owing to Edward Jenner, the famous scientist considered the “father of immunization”, their immunity has been responsible for the discovery of a vaccine used for the prevention of smallpox in humans [1]. Smallpox and monkeypox are closely related; in fact, symptoms such as skin rashes and fever are surprisingly similar. Monkeypox infection was discovered in 1970 in the Democratic Republic of Congo, although nowadays its epidemiology it is still unknown. After the first cases, several infections were also denoted in other continents including America, Australia, Asia, Europe, and the Middle East. Viruses that usually infect animals are confined to their own world. However, as in the case of the pandemic triggered by the SARS-CoV-2 virus, the transmission of the infection due to the Monkeypox virus can evolve and adapt to the environment in a way that allows a “species jump”, starting with animals and then infecting humans [2]. This critical issue is mostly attributable to the promiscuity which may occur between the human and animal worlds, especially in some underdeveloped countries. In the case of monkeypox infection, before the declaration of some cases of monkeypox in the United States of America, cases were endemic only in African countries such as Cameroon, Central African Republic, Democratic Republic of Congo, and Nigeria [3]. It is noteworthy that smallpox immunization with the vaccine has also been effective against monkeypox. However, it has been strongly discouraged to carry out such vaccination since 1980 because this infection shares several clinical and pathological factors with other microbial infections, such as varicella-zoster virus, cowpox, and many others [4]. The current re-emergence of monkeypox cases may be due to both a decrease in vaccination and to a gene development of the virus itself which would be able to directly tackle the human genome. As a result, this aspect should be considered an important cue either to highlight host preferences and possible reservoirs which continue to develop or to continue virus gene sequencing [5].

Compared with COVID-19 infection, Monkeypox expresses a lower incidence of severe forms, even though there are several concerns about the geographical spread and further resurgence of the infection [6,7]. Since 2003, cases of monkeypox have been reported in countries where the disease was not endemic. After 40 years of silencing infections, between 2010 and 2019, a few cases were reported in Nigeria. In particular, it has been reported that transmission occurred after the importation of rodents from Ghana to the United States [8]. It is conceivable that the infection was transmitted from animal to human. The species jump would have generated an epidemic of about 47 definite cases [9]. To date, monkeypox has been reported in Nigerian adults traveling to different countries such as Israel [10], the United Kingdom [11,12], Singapore, and the United States. In the United Kingdom, there has been one case of hospital transmission and two other cases of transmission between family members [13]. As a result, these cases expressed in the United Kingdom led to additional cases of infection: therefore, epidemiological data can demonstrate how infected cases may be carried by travelers [14,15]. It should also be noted that the cases found in the U.K. in 2021 occurred when the monkeypox infection status in Nigeria was low (32 suspected cases of the disease) [14]. The epidemic potential R0 > 1 is equivalent to the result of transmission among humans [15].

The outbreak of a high proportion of monkeypox cases starting in May 2022 with undirected transmission from endemic countries raised several questions about how this infection might have spread and about future expectations. In addition, it is to be speculated that the type of transmission may have changed in the interim. Following the coronavirus pandemic, the specter of a monkeypox pandemic is certainly one of the major issues to be addressed, as it could potentially cause serious health problems. Although compared with the COVID-19 pandemic, it is noteworthy how medicines and prevention tools such as vaccines are ready for use, which is a substantial advantage [16,17].

There are merits and limitations to this report. The strengths are certainly the data that emerged from more than 60 relevant articles, which were used without language and time constraints, allowing a significant reduction in selection bias. However, there are also limitations: first, the fact that many countries where this virus is still active have non-state-of-the-art health care systems; second, the paucity of data on the age of confirmed cases. It follows, therefore, that the number of reported cases may not exactly approximate reality, potentially generating bias in the data which could provide incorrect estimates of contagiousness with reference to the age groups of the infected population. According to some analyses, the median age of infection would be about 35 years, demonstrating an increase over time in the median age of onset, attributable perhaps to a genetic modification of the virus itself [18].

## 2. Results

### Antiviral Drugs

To date, there are no antiviral drugs that have been developed specifically against monkeypox infection. However, there are antiviral agents available to the healthcare community which were originally placed on the market with other therapeutic indications. To date, these agents have obtained or are obtaining the extension of the indication for the treatment of monkeypox. In general, most people diagnosed with monkeypox recover without any treatment; in some cases, symptomatic and supportive treatment is implemented. People who develop severe disease or have immune system impairment may be prescribed an antiviral agent known as Tecovirimat. This medicinal product was originally developed for smallpox and acts by interfering with a protein on the surface of *orthopoxviruses*, the VP37 protein, preventing virion release and thus preventing viruses from reproducing normally, slowing infection spread [19]. In particular, Tecovirimat blocks the interaction of VP37 with GTPase, Rab9, and TIP47 cells, thereby preventing the formation of fully developed cells which are then able to exit the cell and proceed to spread the virus from cell to cell and over long distances. Tecovirimat is administered orally. It was placed on the American market by the FDA and subsequently, based on data from animal and human studies, Tecomirivat was also approved under “exceptional circumstances” by the European Commission for the treatment of *orthopoxvirus* infections (smallpox, monkeypox, and cowpox), with the indication in adults and children with a body weight of at least 13 kg [19,20,21]. In particular, on 13 July 2018, Tecovirimat was approved for marketing in the U.S. market as the first medicine indicated for the treatment of smallpox, to which the indication was later extended to the treatment of MPX [19]. Subsequently, Tecovirimat was also authorized for marketing in the European Union on 6 January 2022 [20]. In fact, in the context of the monkeypox epidemic, the EU government has already facilitated large multinational studies in the EU on the use of antiviral Tecovirimat by reviewing testing protocols and collaborating with the Clinical Trial Coordination Group (CTCG), acting on national regulations in collaboration with the boards which coordinate and can facilitate the approval of clinical trial applications by the competent national authorities. However, the drug is not widely available in Europe, and as a result, treatment with Tecovirimat should only be considered under investigational or compassionate use protocols, particularly for patients who have severe disease or who may be at risk of developing complications, such as immunocompromised people. For patients at high risk of progression to severe disease, treatment should be given early in the course of the disease along with supportive care and pain control. Other medicinal products under study include cidofovir and brincidofovir, which are two antiviral agents administered by infusion. Cidofovir is an antiviral drug approved in both the United States and the European Union for the treatment of cytomegalovirus (CMV) retinitis in patients with acquired immunodeficiency syndrome (AIDS) [22]. No data from RCT are available on the efficacy of cidofovir in treating human cases of monkeypox, although in real-life studies all patients had a complete recovery [23,24]. Furthermore, it has been shown to be effective against *orthopoxviruses* in in vitro and animal studies [25]. Cidofovir has significant nephrotoxicity that limits its use as a first-line treatment. Brincidofovir, on the other hand, is a prodrug of the antiviral cidofovir [26]. There is lipid conjugation with cidofovir, thus allowing this drug to be used at lower concentrations, consequently reducing its toxicity and still allowing for targeted action on the inhibition of viable DNA replication. Brincidofovir is already on the American market for the treatment of cytomegalovirus and is currently approved for the treatment of smallpox in adults and pediatric patients, including children. As for cidofovir, there are no data to support its efficacy in the treatment of monkeypox [25,26]; however, in vitro and animal studies demonstrated efficacy for the treatment of *orthopoxvirus* [27]. To date, cidofovir has not obtained an extension of the indication for the treatment of monkeypox infection in the EU. Brincidofovir is not currently authorized in the EU. Limited studies of drug use in some cases of monkeypox have shown that Tecovirimat is more effective than brincidofovir because the latter can develop pharmacoresistance with mutations in F13L (highly conserved viral membrane protein) and E9L (DNA polymerase) [28,29,30].

## 3. Discussion

Since the beginning of the monkeypox epidemic and until 1 November 2022, there have been approximately 20,000 confirmed cases of monkeypox (MPX) in EU/EEA countries. The EU/EEA countries reporting the highest number of confirmed cases are Spain, France, and Germany. The current ongoing MPX epidemic appears to be caused by the Clade II b variant of the monkeypox virus, which is clinically less severe, has lower interhuman transmissibility, and lower lethality. The degree of individual response to the disease depends on the person’s immune response. To date, there is no approved medicinal treatment specifically for monkeypox infections. However, some antivirals developed for use in smallpox patients may prove useful against MPX: Tecovirimat or ST-246, cidofovir, and brincidofovir. The spread of this epidemic with different characteristics from previous ones and in places rarely called upon to handle monkeypox has increased attention to what until a few weeks ago might have been considered a neglected disease in its own right: however, efficacy data from human studies for these three active ingredients with antiviral action are lacking to date. In addition, there are no available specific vaccines developed to prevent MPX eradication and to protect against MPX infection, although smallpox vaccines could also provide immunity against MPX infection. Eventually, intravenous vaccinia immune globulin (VIGIV), which is licensed for the treatment of complications from smallpox (vaccinia) vaccination, may be authorized for use to treat monkeypox and other pox viruses during an outbreak. So far, data have been provided only from studies conducted in vitro and in animals; thus, clinical evidence on efficacy against MPX is needed for antiviral agents. To date, there are few results demonstrating antiviral activity against *orthopoxvirus*. Tecovirimat was found to provide higher survival rates than placebo, whereas cidofovir and brincidofovir in the treatment of monkeypox cases in people have no supporting clinical evidence, only data of antiviral activity on *orthopoxvirus*. In addition, no data are available on the efficacy of VIGIV in the treatment of monkeypox virus infection. The use of VIGIV has no proven benefit in the treatment of monkeypox, and it is not known whether a person with severe monkeypox infection would benefit from treatment with VIGIV. All patients treated with Tecovirimat recovered from monkeypox, expressing only one notable adverse event. All patients treated with brincidofovir recovered from monkeypox, but all expressed an increase in alanine transaminase. Other adverse events include nausea and abdominal discomfort. To date, no studies have been conducted in patients treated with cidofovir. Tecovirimat may be the best treatment being administered and taken orally and demonstrating fewer adverse events. However, it is noteworthy that the efficacy and adverse events of these antiviral agents have been evaluated in a very limited number of patients and the limited evidence currently available confirms that further studies are needed to evaluate their effectiveness and safety as useful treatments for human monkeypox [23,24]. There are currently no active substances indicated for the treatment of monkeypox infection in humans, and in all likelihood there will not be any for several months, given the time required for the development of new antivirals and a need that will hardly become as pressing as it did in the case of COVID-19. It is currently not possible to predict how this wave of infections will evolve, and there are no preventive or curative tools specifically indicated for the treatment of monkeypox. On the one hand, it is therefore essential to develop well-organized clinical trials and real-world studies to generate the right evidence from the antiviral medicinal products currently on the market in order to better determine the clinical use of these drugs and to provide more evidence of efficacy and safety. On the other hand, antiviral agents that we currently have at our disposal are useful tools but may not be enough as an armamentarium to counteract a possible worsening of the epidemiological situation of this virus. As a result, it is necessary to further study the structure of the virus while monitoring the epidemiological situation and trying to develop specific new agents as an effective means to counter the development of infection. Vaccines themselves as preventive tools should be studied more and developed to protect and ensure public health defense.

## 4. Materials and Methods

### Biology and Replication

Monkeypox virus (MPXV) belongs to the large enveloped virus family *Poxviridae* [31]. Poxviruses are large, enveloped viruses. Their genome has double-stranded linear DNA (dsDNA) made up of around 200 kilobase pairs to form around 200 genes. Half of the genes are present in many vertebrate poxviruses and serve for viral replication, while the remaining half of the genes are what are called accessory genes which serve for virus–host interactions and are not important for viral replication [32]. The family of these viruses has over ten types including vaccinia virus (VV), variola virus (VARV), cherry mottle leaf virus (CMLV), cowpox virus (CPXV), and several new variants isolated from infected humans since 2010 [33,34,35]. However, it appears that all species are descended from a rodent virus, including MPXV and CPXV, which use mice as hosts for the viral infection reservoir [35]. The diversity of hosts and their virulence are the main strengths: MPXV, CPXV, and VACV can thus infect and transmit between many types of mammals. Smallpox virus can cause smallpox disease with a high mortality rate of up to 30%. Human monkeypox is a typical zoonosis which clinically resembles smallpox: MPX shows a lower human-to-human transmission with reduced mortality rate [36,37].

Transmission occurs through biological fluids, respiratory droplets, and wound material. As a virus it is quite stable, which is evidenced by the fact that studied scabs cultured for 13 years in the laboratory remained unchanged [37]. Monkeypoxvirus (MPXV), in spite of its name, is widespread in African rodents and particularly in squirrels, which are now considered the virus’s maintenance reservoir [38,39]. To date, two strains of MPXV are known: a more virulent one that can lead to 10% mortality in the Congo Basin, while another milder one is found in West Africa [40]. MPXV can still infect other animals such as dogs and other rodents at relatively low doses [41,42,43,44,45,46]. “Wild mice” are susceptible to MPXV (CAST/eij strain of mice) and differ from the more resistant classical mice [47,48,49,50,51]. The CAST mouse/MPXV model may have advantages for studying correlations of immunity and vaccine efficacy. Monkeypox has similar but milder features than smallpox. Its manifestation consists of three stages:Incubation: can vary from 7 to 14 days, but is generally about 13 days.Prodromal phase: includes fever and lymphadenopathy.Skin rash.

Lymphadenopathy characteristic of the prodromal phase is the essential element which distinguishes monkeypox from smallpox and chickenpox.

The rash also deserves a separate characterization: it is in turn characterized in several phases. In an initial macular phase, the papules appear rosaceous, flat, and not raised. These papules then become denser, vesicular, and pustular. They later evolve further to become scabs that will then inevitably fall off. The rash can affect the face, trunk, and extremities, and sometimes the genitals, and all of these areas are involved at the same level, so the manifestation occurs simultaneously in all of the above areas. Extreme care must be taken with papules in the pustular phase, as they contain active viruses that by direct transmission can infect another individual [52]. Secondary symptomatology can be of serious concern compared with primary manifestations. This, in fact, can occur with diseases such as bronchopneumonia, gastroenteritis, sepsis, encephalitis, and keratitis [53]. Although it is not yet known how monkeypox manages to circulate in the wild, in recent decades the research world has increasingly studied the strategy to prevent this same virus from infecting new hosts outside endemic areas. Certainly, one of the main reasons why this virus is able to remain active in the wild is that it uses different hosts. The implantation of this virus in ground squirrels in regions of North America is arousing attention and alertness in the scientific world, as for the foreseeable future there are fears of a potential outbreak that could be very dangerous [54]. The fact that MPX can conceal itself in any mammalian cell with a variable time interval in which it remains in the host makes this virus potentially dangerous. Infectivity can depend on several factors, first of all the reactivity of the host’s immune system. Antiviral factor K is a protein that inhibits the multiplication of the viral genetic material and, consequently, the multiplication of the virus, thus causing the infection to stop or be attenuated. It is targeted by two viral genes: E3L and K3L. It has been found that primate K protein genes have undergone substantial modifications and are susceptible to inhibition of the K3L gene [55,56]. The main change is due to the viral genome, which has developed small inhibitors of the human antiviral protein K. For these reasons, it is very likely that much more virulent variants could be generated in humans. Actually, the greatest fear is that other types of variants of the same virus family could be generated. Smallpox virus mutates less rapidly than RNA viruses because the DNA genome has a DNA polymerase that has a 3′-5′ exonuclease correction activity [57]. The mutation rate of poxviruses could result in up to two nucleotide mutations in the genome per year compared with the dozens that can occur in an RNA virus [58,59,60]. Poxviruses therefore vary much less than SARS-CoV-2; however, it must be said that the smallpox virus genome is large and flexible and allows for large changes to the structure causing loss or increase in genes and thus altering viral phenotypes very rapidly [61]. Generally, however, mutants have repeats in their genome of a viral gene that is often the direct target of the drug therapy being undertaken [62,63] and for this reason the virus, in order to attempt to resist the pharmacological effect, aims to increase the production of that affected gene as a survival mechanism.

## 5. Conclusions

The epidemic caused by monkeypox, unlike COVID-19, is not having the same severity and speed of spread, both because of the different biological characteristics of the virus and the ready availability of different vaccines and antiviral agents for smallpox and monkeypox. The rapid initiation of infection control measures and the use of vaccines and antiviral agents are important strategies for controlling the monkeypox epidemic. Regardless, the new reality is that human monkeypox is no longer a rare zoonotic disease and needs more public health attention. To limit as much as possible a new scenario that could resemble the recent pandemic from which we may be about to emerge, the biology of this virus certainly needs to be studied in depth to try to assess any genetic changes while keeping them under control and limiting transmission to humans as much as possible.

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
