# Peer review of "Pharmacological Agents with Antiviral Activity against Monkeypox Infection"

_ijms, 2022, doi:10.3390/ijms232415941_

Round 1

Reviewer 1 Report

In this article, the author discussed the epidemic of monkeypox and antiviral therapeutics against the virus. The article is well-organized and provided a comprehensive view of monkeypox infection, epidemiology, and antiviral therapies. The discussion of current antivirals and vaccines against the monkeypox virus gives insight into the future development of drugs and control of viral infection. However, there are some issues need to be addressed to improve the manuscript:

1.     I suggest the authors go over the manuscript again carefully since there are many mistakes in the manuscript. For example, the title “monkeypox infectous” should be corrected.

2.     In the abstract, the first sentence “caused by a virus a member of the genus” should be corrected. “enlarged lymphnodes New vaccines” coma missing.

3.     In the first part, check the first 7 lines to correct mistakes.

4.     First part, paragraph 2 “there have been reports which transmission would have occurred as a result of the impor-“ is confusing

5.     Part 2 “heir genome has double-stranded linear DNA (dsDNA) made up of around 200 kilobase pairs to form around 200 genes” correct the sentence.

6.     Part 2 “cowpox virus (VACV), cowpox virus (VARV), camel-pox virus (CMLV), cowpox virus (CPXV)” should spell out the full name of the virus.

7.     Part 2 last sentence “and because of this the virus in order to resist the effect of the drug in-creases the production of that affected gene” need to correct, it is confusing.

8.     Part 3 “but there are available to the health community only anti-viral agents authorized for the marketing” is confusing.

9.     Part 4 “Spain, France Germany” need to correct. “MXP” needs to correct.

Author Response

Reviewer 1

In this article, the author discussed the epidemic of monkeypox and antiviral therapeutics against the virus. The article is well-organized and provided a comprehensive view of monkeypox infection, epidemiology, and antiviral therapies. The discussion of current antivirals and vaccines against the monkeypox virus gives insight into the future development of drugs and control of viral infection. However, there are some issues need to be addressed to improve the manuscript:

  1. I suggest the authors go over the manuscript again carefully since there are many mistakes in the manuscript. For example, the title “monkeypox infectous” should be corrected. Accepted and amended.
  2. In the abstract, the first sentence “caused by a virus a member of the genus” should be corrected. “enlarged lymphnodes New vaccines” coma missing. Accepted and amended.
  3. In the first part, check the first 7 lines to correct mistakes. Accepted and amended.
  4. First part, paragraph 2 “there have been reports which transmission would have occurred as a result of the impor-“ is confusing. Accepted and amended.
  5. Part 2 “heir genome has double-stranded linear DNA (dsDNA) made up of around 200 kilobase pairs to form around 200 genes” correct the sentence. Accepted and amended.
  6. Part 2 “cowpox virus (VACV), cowpox virus (VARV), camel-pox virus (CMLV), cowpox virus (CPXV)” should spell out the full name of the virus. Accepted and amended.
  7. Part 2 last sentence “and because of this the virus in order to resist the effect of the drug in-creases the production of that affected gene” need to correct, it is confusing. Accepted and amended.
  8. Part 3 “but there are available to the health community only anti-viral agents authorized for the marketing” is confusing. Accepted and amended.
  9. Part 4 “Spain, France Germany” need to correct. “MXP” needs to correct. Accepted and amended.

Reviewer 2 Report

Thank you for sharing the interesting article. Here some edits and comments that could help to improve the article.

Section 1

-"but nowadays epidemiology it is unknown"; please rephrase for more clarity

-regarding "other countries" mentioned, "Australasia" seems incorrect; also Europe and the Middle East are no individual countries but comprise several countries so please correct

-"occurs" should possibly read as "occur"

-"due both to [...] to a" should be rephrased for clarity

-"affect human genome" needs clarification; do you mean that the virus directly tackles the human genome?

-not clear what you mean by "which transmission would have occurred as a result of the importation of rodents from Ghana to the United States"; also please ad the related reference

-also not sure what you mean by "monkeypox appears to have been detected in Nigerian adults in several countries such as"

-same, unclear what "carried out to additional cases of infection [...] travellers can be considered potential outbreaks of infection transmission" means

-"line of transmission" seems an unusual term

-likewise " medicines and prevention weapons are previously ready" isn't clear either

-"generating biases in the data that could provide an estimate of contagiousness on population age groups" needs rewording

-please support the analyses you are referring to in the last sentence of this section with an appropriate reference

Section 2

-correct "heir genome" to "Their genome"

-regarding "descended from a rodent", do you mean "descended from a rodent virus"?

-not clear what you are aiming to state by "studied scabs cultured for 13 years in the laboratory remained unchanged"; was scab truly cultured for 13 years and what remained unchanged?

-consider using another term than "chief" and rephrase "chief among them  the reactivity of the host's immune system"

-please reword "the virus in order to resist the effect of the drug increase the production of that affected gene" for more clarity

Section 3

-Please rewrite and shorten the 1st sentence of this section "At present there are [...] treatment of monkeypox"

-Please check throughout whether the drug is spelled "Tecovirimat" or "Tecomirivat" or "tecovirimat" or are those two different drugs?

-consider using another terms than "weapons" in the context of vaccines and antiviral agents 

Author Response

Thank you for sharing the interesting article. Here some edits and comments that could help to improve the article.

Section 1

-"but nowadays epidemiology it is unknown"; please rephrase for more clarity. Accepted and amended.

-regarding "other countries" mentioned, "Australasia" seems incorrect; also Europe and the Middle East are no individual countries but comprise several countries so please correct. Accepted and amended.

-"occurs" should possibly read as "occur". Accepted and amended.

-"due both to [...] to a" should be rephrased for clarity. Accepted and amended.

-"affect human genome" needs clarification; do you mean that the virus directly tackles the human genome? We have more clarified this point as suggested.

-not clear what you mean by "which transmission would have occurred as a result of the importation of rodents from Ghana to the United States"; also please ad the related reference. We added a specific reference.

-also not sure what you mean by "monkeypox appears to have been detected in Nigerian adults in several countries such as". We have more clarified this point as suggested.

-same, unclear what "carried out to additional cases of infection [...] travellers can be considered potential outbreaks of infection transmission" means. We have more clarified this point as suggested adding specific references.

-"line of transmission" seems an unusual term. Accepted and amended.

-likewise " medicines and prevention weapons are previously ready" isn't clear either. We have more clarified this point as suggested.

-"generating biases in the data that could provide an estimate of contagiousness on population age groups" needs rewording. Accepted and reworded.

-please support the analyses you are referring to in the last sentence of this section with an appropriate reference. We added a specific reference.

Section 2

-correct "heir genome" to "Their genome". Accepted and amended.

-regarding "descended from a rodent", do you mean "descended from a rodent virus"? We have more clarified this point adding a specific reference.

-not clear what you are aiming to state by "studied scabs cultured for 13 years in the laboratory remained unchanged"; was scab truly cultured for 13 years and what remained unchanged? Yes, it is, there is a specific reference to this sentence that describes the number of particles per scab after 13 years of culture.

-consider using another term than "chief" and rephrase "chief among them the reactivity of the host's immune system". Accepted and reworded.

-please reword "the virus in order to resist the effect of the drug increase the production of that affected gene" for more clarity. Accepted and reworded.

Section 3

-Please rewrite and shorten the 1st sentence of this section "At present there are [...] treatment of monkeypox". Accepted and reworded.

-Please check throughout whether the drug is spelled "Tecovirimat" or "Tecomirivat" or "tecovirimat" or are those two different drugs? Accepted and amended. The name is Tecovirimat.

-consider using another terms than "weapons" in the context of vaccines and antiviral agents. Accepted and amended with “tools”.

Reviewer 3 Report

This study provided sufficient experimental evidence to support the conclusion.

Overall the results are solid and consistent. Several issues still need to be addressed before publication. The following questions may help the authors to improve the quality of their already outstanding manuscript.

In many places, the flow is interrupted due to grammatical errors. Redundancy of text and too wordy text. In many places, the meaning of sentences is not clear. Authors are advised to carefully go through the manuscript and make it crisp and make other corrections where ever needed

Author Response

This study provided sufficient experimental evidence to support the conclusion.

Overall the results are solid and consistent. Several issues still need to be addressed before publication. The following questions may help the authors to improve the quality of their already outstanding manuscript.

In many places, the flow is interrupted due to grammatical errors. Redundancy of text and too wordy text. In many places, the meaning of sentences is not clear. Authors are advised to carefully go through the manuscript and make it crisp and make other corrections where ever needed.

Thank you for the suggestions. We followed all the reviewers' advices and we implemented all the amendments requested. We also reworded several sentences and deleted grammars mistakes. We hope the manuscript is now more fluent and clear.

Round 2

Reviewer 2 Report

All comments and suggested edits were addressed sufficiently.

Reviewer 3 Report

accept